# LIFEPLAN: A worldwide biodiversity sampling design

Bess Hardwick [1] [⍥], Deirdre Kerdraon [2] [⍥] *, Hanna M. K. Rogers[2],
Dimby Raharinjanahary[3], Eric Tsiriniaina Rajoelison[3], Tommi Mononen [1],
Petteri Lehikoinen[4], Gaia Banelyte[2], Arielle Farrell[2], Brian L. Fisher[3,5], Tomas Roslin[1,2],
Otso Ovaskainen[1,6]

1 Organismal and Evolutionary Biology Research Programme, University of Helsinki, Helsinki, Finland,
2 Department of Ecology, Swedish University of Agricultural Sciences, Uppsala, Sweden, 3 Madagascar Biodiversity Center, Antananarivo, Madagascar, 4 The Finnish Museum of Natural History, University of Helsinki, Helsinki, Finland, 5 California Academy of Sciences, San Francisco, CA, United States of America, 6 Department of Biological and Environmental Science, University of Jyväskylä, Jyväskylä, Finland

⍥ These authors contributed equally to this work.
* deirdre.kerdraon@slu.se

**Data Availability Statement:** No datasets were generated or analyzed during the current study. Unrestricted data from this study will be made available upon study completion.

## Abstract

As the technology for mass identification of species is advancing rapidly, we developed a field sampling method that takes advantage of the emerging possibilities of combining sensor-based data with automated high-throughput data processing. This article describes the five field sampling methods used by the LIFEPLAN project to collect biodiversity data in a systematic manner, all over the world. These methods are designed for use by anyone with basic biology or ecology knowledge from the higher education or university level. We present the selection and characteristics of international sampling locations for urban and natural sites, as well as the nested scale design in the Nordic countries and Madagascar. We describe the steps to collect sequences of animal images (.jpg) from infrared triggered camera traps, audio data (.WAV) of environment sounds from audio recorders, invertebrate samples in ethanol from Malaise traps for DNA metabarcoding, as well as both soil samples and 24-hour air samples obtained from cyclone samplers for fungal DNA metabarcoding. To ensure the usability and consistency of the data for future use, we pay particular attention to the metadata collected. In specifying the current sampling protocols, we note that technology will continue to improve and evolve. Hardware will also change within a short time period, with the advantage of improving the equipment used for collecting samples. Thus, we present examples of the samples collected by each current sampling method, to be used as a baseline or in comparison with different equipment models.

## Introduction

We have so far only discovered an estimated 20% of all species on planet Earth [1]; at the same time, species are going extinct at an alarming rate [2]. If we are to understand current species diversity and predict future scenarios in a rapidly changing environment, monitoring

**Funding:** OO, TR: grant number 856506. European Research Council (ERC), https://erc.europa.eu/homepage. The funder did not play any role in the study design, data collection and analysis, decision to publish or preparation of the manuscript. The work of all the authors was supported by the same grant awarded to OO and TR.

**Competing interests:** The authors have declared that no competing interests exist.

biodiversity over extended time-periods, across a wide variety of different habitats, and in all major species groups, is essential. To establish a baseline against which to gauge future change, this monitoring needs to start now.

Current biodiversity surveys are often limited in spatial coverage and timespan, biased towards regions with the most people and financial resources, and restricted to species that are easiest to see and identify [3–6]. As experts using morphological identification are commonly specialists in one or a few groups, there are simply not enough experts to produce biodiversity data with truly global coverage of the most diverse species groups [7]. In recent years, the development of molecular identification and machine learning methods, as well as increases in computing power, have enabled us to teach algorithms to identify species [8–10]. The sheer number of samples that can be identified by a computer is thus far greater than what human experts could ever process.

To take full advantage of these new methods, we have devised sampling designs that generate consistent data. In the LIFEPLAN project, sampling started in 2021 and is to continue until the end of 2025. A central aim was to create a standardised sampling scheme for monitoring terrestrial animal and fungal biodiversity under any climatic conditions. These methods require minimal technical or taxonomic expertise, allowing a wide range of participants to collect data. While many national long term monitoring endeavours already exist [11–13], LIFE-PLAN is the first to create a worldwide sampling design to systematically collect data to target a broad range of taxonomic groups, which can be used for monitoring biodiversity throughout the year. By combining these sampling methods on a large scale with automated species identification and novel statistical methods, LIFEPLAN aims to generate a predictive understanding of global biodiversity. These methods are replicable in time and space, which allows any future dataset that would be collected by identical methods to be compared to the original Lifeplan data. While the technology will improve, we will still be able to infer changes in biodiversity by matching the sampling design.

The sampling design of LIFEPLAN is aimed at covering major terrestrial taxa excluding plants; thus, sampling is aimed at arthropods, fungi and vertebrates. Arthropods are collected with passive Malaise traps. To learn both what fungi are dispersing in the area as well as which species are actually growing in the soil, we collect fungal spores from the air with cyclone samplers as well as soil fungi from soil samples. For large or audible groups such as mammals and birds, we use camera traps and audio recorders, which are non-invasive and low-cost.

Even these semi-automated methods do require regular personnel to collect the samples every week year-round. The personnel must be able to negotiate the common challenges of field work in their local conditions and have basic computer skills. Some financial resources are also required to purchase the specialised equipment and consumables, and to fund the DNA sequencing. Having all these elements in place at a site that is also suitable for sampling has limited global LIFEPLAN coverage, as such sites are much easier to find in regions with more financial resources.

The specific makes and models of equipment used have had to be cost-effective, but are challenged by some climatic conditions. Humidity is causing breakage in camera traps and AudioMoths, and cyclone samplers are struggling in freezing temperatures. International shipping of physical samples to a DNA sequencing facility requires special permits, and transferring the large amounts of image and audio data collected requires a stable and reasonably fast internet connection.

The methods for identifying species from these samples are still developing and improving. One limitation is the availability of annotated reference data [11, 14–21] that connects a DNA barcode sequence, image or sound to a known species. We are unable to identify all the species collected, and many of the species that we collect are novel and require taxonomical work to

even describe them. Yet, now is the time to collect an archive of samples, as the species we are looking at might be lost by the time we can identify them. This is not to say that species devoid of a traditional scientific binomial could not contribute to present-day analyses. Of the species detected but not yet assigned a traditional scientific name, we can form species hypotheses and thus efficiently estimate both community-level attributes such as species numbers, as well as spatial and temporal turnover in community composition [22–25]. We can also characterise phylogenetic and functional diversity at the community level [26–30]. And for the interim taxa delimited, we can characterise distributions and their drivers [31].

In terms of the focal ecosystems of LIFEPLAN, we have clearly targeted terrestrial biodiversity. Since any realistic sampling scheme will have to select some targets at the expense of others, we have chosen not to sample plants, or ground or tree crown-dwelling invertebrates. Furthermore, the LIFEPLAN project is not yet using the data collected to its full capacity due to funding and time constraints. More data can be acquired from the collected samples as technology improves. Spore samples are likely also to include pollen, the DNA of which could be sequenced as well, and soil samples will also contain much more than fungi. More groups of vocalising taxa can be identified from audio recordings (such as amphibians and insects).The sampling sites can also be fitted with further sampling devices, such as sampling for aerosol-borne DNA [32]. Current initiatives include adding site-specific measurements of ecosystem processes.

## Materials and methods

The protocols described in this peer-reviewed article are published on protocols.io, updated March 6, 2024, and are included for printing as S1–S5 Files with this article.

**Camera.** The protocol described in this peer-reviewed article is published on protocols.io, dx.doi.org/10.17504/protocols.io.q26g7pxp1gwz/v3 and is included for printing as S1 File with this article.

**Audio.** The protocol described in this peer-reviewed article is published on protocols.io, **dx.doi.org/10.17504/protocols.io.kqdg3xbp1g25/v2** and is included for printing as S2 File with this article.

**Cyclone sampler.** The protocol described in this peer-reviewed article is published on protocols.io, dx.doi.org/10.17504/protocols.io.6qpvr3zn2vmk/v1 and is included for printing as S3 File with this article.

**Malaise trap.** The protocol described in this peer-reviewed article is published on protocols.io, dx.doi.org/10.17504/protocols.io.kqdg3xkdqg25/v2 and is included for printing as S4 File with this article.

**Soil sampling.** The protocol described in this peer-reviewed article is published on protocols.io, dx.doi.org/10.17504/protocols.io.5jyl8pw9rg2w/v2 and is included for printing as S5 File with this article.

## Site selection

Since biodiversity patterns may be differentially impacted by different community assembly processes at different spatial scales [33, 34], we opted for a hierarchical sampling design, spanning six orders of magnitude in spatial scales (from $10^{-1}$ km to $10^4$ km).

The Global ($10^4$ km scale) sampling consisted of 73 sampling stations spread across all continents. Sampling at the National ($10^3$ km) scale was based on 43 sampling stations, and at the Nested ($10^2$–$10^{-1}$ km) scale on another 43 sampling stations. Sampling at the National and Nested scales focused on two contrasting locations: the Nordic countries (Finland, Norway, Sweden) and Madagascar. Given their widely different patterns of climatic conditions and

regional and local endemism [35–37], this regional contrast yields insight into the context-dependency of the results achieved–as then validated by global patterns.

We chose the Nordic countries and Madagascar for National and Nested scale sampling because they are of comparable area but have very different evolutionary histories. The Nordic countries were colonised after the most recent ice age, while in Madagascar, species have been evolving in isolation for millions of years [35]. Nordic species are very well known, while the enormous biodiversity of Madagascar is still poorly understood [35, 37–39]. The Swedish insect fauna consists of postglacial immigrants and is one of the best known in the world. Madagascar is the world's most important biodiversity hotspot, with a long period of evolution in isolation–as combined with recent, massive deforestation [40]. Many species are critically endangered if not already extinct [41, 42]. By comparison, endemic species in the Nordic countries are next to absent. Given these contrasts, we aim to resolve–among other things—differences in range size between Nordic and Malagasy taxa, patterns of turnover in space and time, and differences in phylogenetic community structure between the respective regions. Beyond providing a general understanding of basic community assembly processes, each of these insights is needed for improved conservation prioritization—thus attesting to the value of large-scale biodiversity survey and biomonitoring programmes.

To resolve patterns of community structure and turnover in space and time, we have invented a hierarchically structured sampling design in Sweden and Madagascar. At the National scale in Madagascar, we have selected 25 sites spanning all major forest types. In the Nordic countries, we selected 18 sites based on their contribution to spatially even coverage of the region. For the Nested scale, we chose an area in southern Sweden and another in Madagascar that were accessible enough to be covered by a single sampling team, were covered by forest, and where landowners gave permission to sample.

Overall, we will thus achieve site-by-time-by-species tables of fungi as airborne spores and mycelia in soil, insects, birds and mammals, across hierarchically Nested scales.

## Global design: Natural vs. urban

The largest spatial scale in our study design is the Global scale. We initially invited researchers from all over the world to submit proposals for study sites. Of 184 sites proposed for the Global design, we selected 87, of which 14 eventually had to drop out (Fig 1).

In the Global design, sites form natural/urban pairs, where the urban location is in or immediately adjacent to a built-up area, and the natural location is in an undisturbed setting with little direct human influence. In other respects, such as altitude and vegetation type, we have aimed to have the two locations be as similar as possible, and separated by 20–50 kilometres (Fig 2). We have tried to cover the world map as evenly as possible, and within the selected regions we have asked teams to sample a habitat that is typical for their region. As a result, we are currently sampling in a wide range of habitats from various forest types to savannah, wetlands, desert and tundra. Teams alternate yearly between their natural and urban locations, so that only one location of each pair is sampled at one time. To avoid confounding the effects of urbanisation and sampling year, we have randomised the design so that half of the teams started in the urban location and the other half started in the natural location.

In practice, levels of urbanisation vary in different parts of the world so that a "natural" location in a densely populated area such as central Europe may be much more urban than an "urban" location in a sparsely populated area such as Canada. There are many different measures of urbanisation, but using e.g. the Human Footprint Index 2009 [43], we see that for most, but not all, site pairs the "urban" location scores higher than the "natural" (Fig 3B). "Natural" locations also often ended up at higher altitudes, as lower-altitude areas were fully built

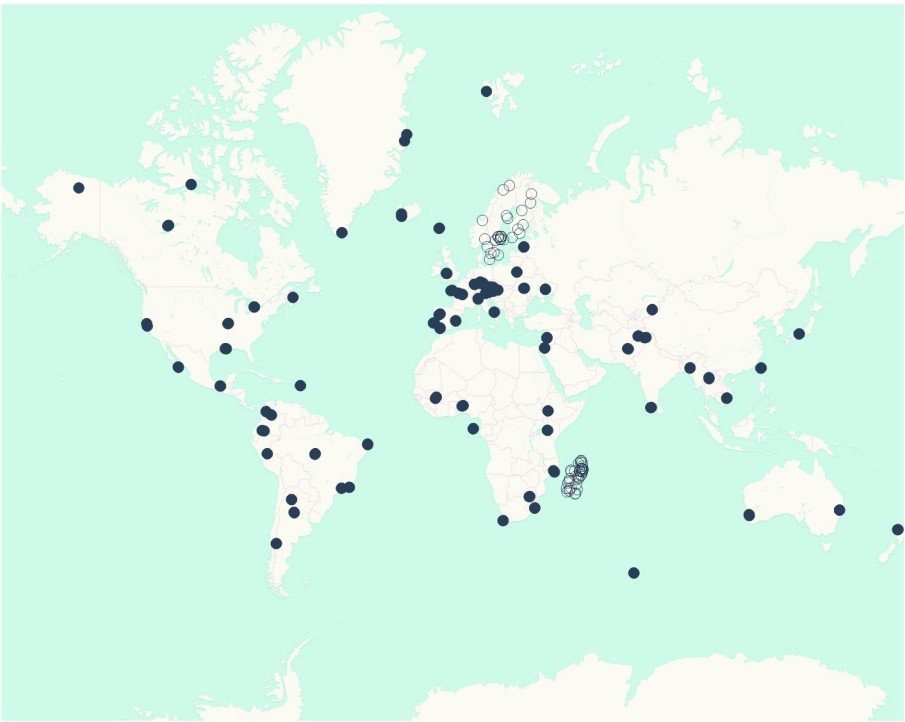

**Fig 1. Locations of LIFEPLAN sites around the world.** Global sites are shown as black dots. The dense clusters of points without fill in Madagascar and the Nordic countries are the sites forming the National and Nested designs.

up in many parts of the world (Fig 3A). Other practical considerations used to restrict site selection included accessibility, security, financial constraints and land ownership.

## National design

The National scale was the intermediate between the Global and Nested scales. At this scale we do not alternate between a natural and urban location, and have instead chosen natural type

### Distances between paired sites

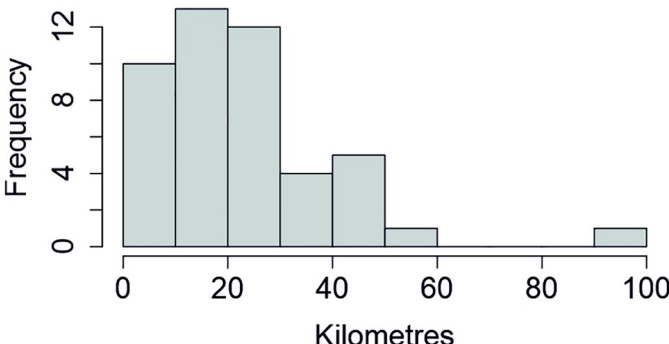

**Fig 2. Histogram of actual distances within site pairs.** In choosing natural / urban site pairs, we aimed for a distance of 20–50 km within pairs. Finding a natural / urban site pair 20–50 km apart that was also accessible and otherwise comparable was not always possible, and we accepted some site pairs that fell outside this range if the locations were otherwise valuable in terms of global coverage. The actual distribution of distances within the finally selected site pairs is shown here.

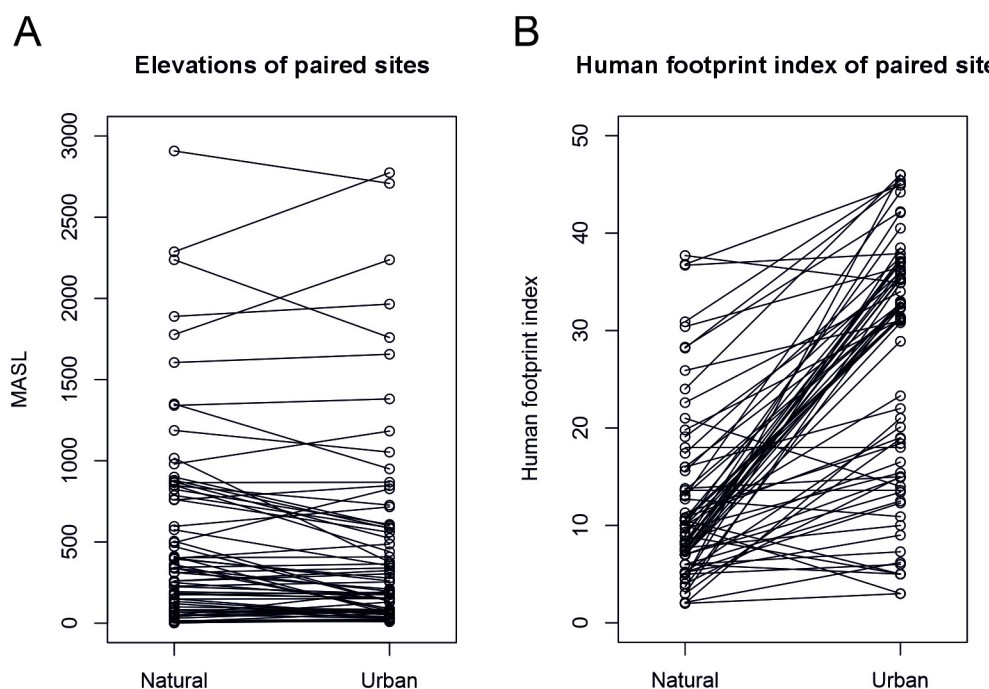

**Fig 3. Comparisons within site pairs.** (A) Differences in elevation (metres above sea level) within natural/urban site pairs. (B) Differences in Human Footprint Index within natural/urban site pairs.

sites that are relatively undisturbed and representative of the local environment. At the National level, we sample in Madagascar as well as the Nordic countries of Sweden, Norway and Finland (Fig 4).

## Nested design

To study hierarchical patterns of community structure in space, we have created a Nested sampling design with increasing distance between sites (Fig 5). We originally searched for sites in a

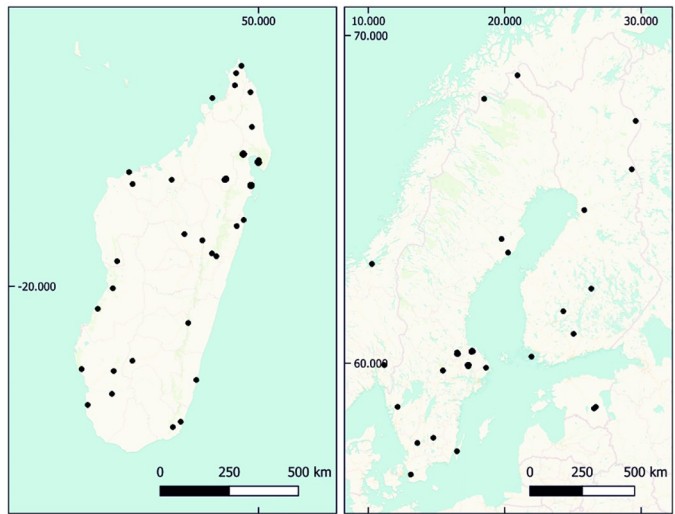

**Fig 4. Locations of sites in the National design.** Black points show National design sites in Madagascar (left) and the Nordic countries (right).

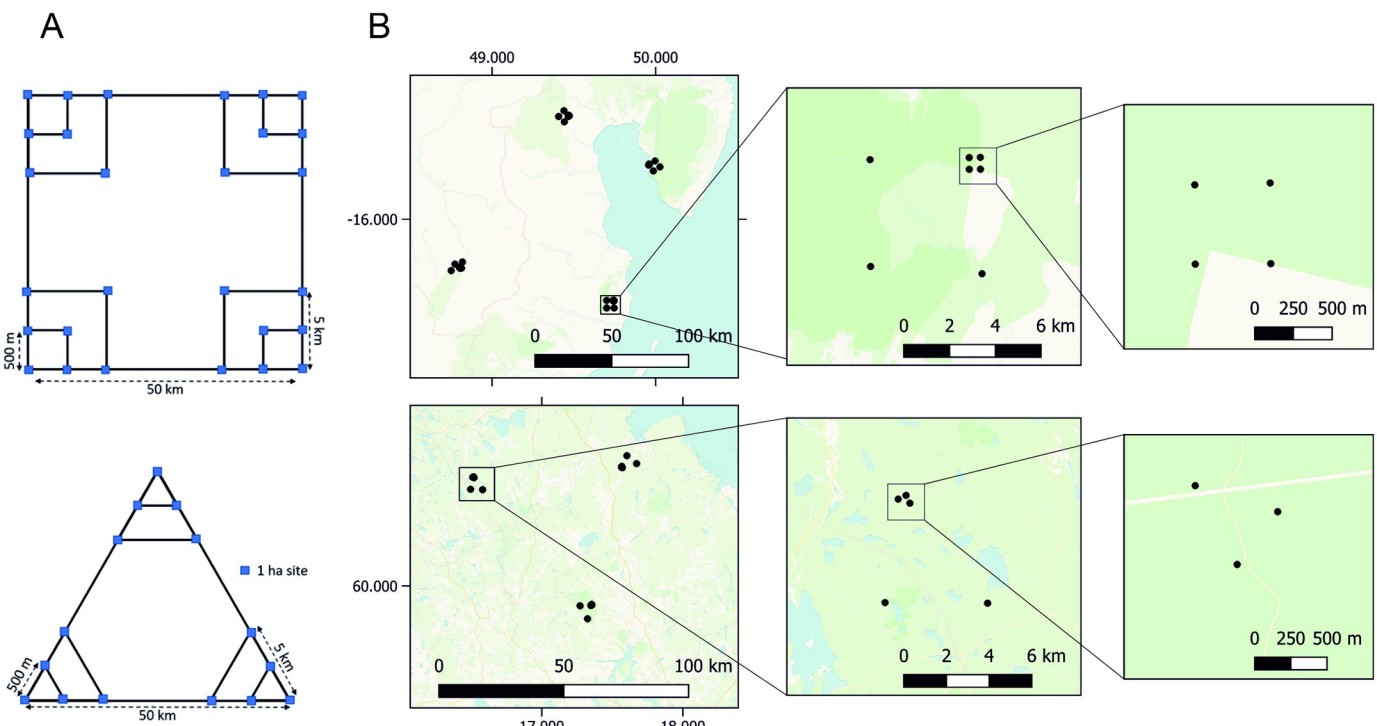

**Fig 5. Locations of sites in the Nested design.** (A) Schematic of the Nested scale field design in Madagascar (top) and Sweden (bottom). (B) Maps showing the actual but approximate site locations for the Nested design in Madagascar (top) and Sweden (bottom).

Nested design that were on the order of 50 km, 5 km and 500 m apart. In Madagascar especially, the search was constrained by the scarcity of remaining forest fragments and the highest-level distances ended up being approximately 70–120 km. In Sweden we achieved highest-level distances of approximately 60–70 km. In Sweden, we implemented a triangular Nested design (n = 3) with 3 clusters each containing 5 sites, and in Madagascar, we had a square design (n = 4) with 4 clusters each containing 7 sites. The triangular design in used in Sweden was reduced from an originally square design due to the high costs of sampling in Sweden. We note that the original square design includes higher coverage of a wider range of pairwise distances, simply by virtue of having more sampling points. Thus, this design was prioritized in Madagascar, as Madagascar has higher beta diversity [37]. If financially feasible, we recommend the square design especially in regions of high beta diversity.

## Site design

The two site designs accommodate different budgets (Fig 6). The extensive design covers a one-hectare square: five cameras and audio recorders are placed in each corner and the centre. Soil samples are collected from each of the same five points. In the simplified design, three cameras and audio recorders are placed along a transect of 140 m, with soil samples collected at each of the three points. The reduced design corresponds to the extensive design with two diagonally opposite corners removed.

We started with the extended design in October 2020 and switched to the reduced design in May 2023. This switch was caused by financial constraints, due to equipment breaking sooner than expected.

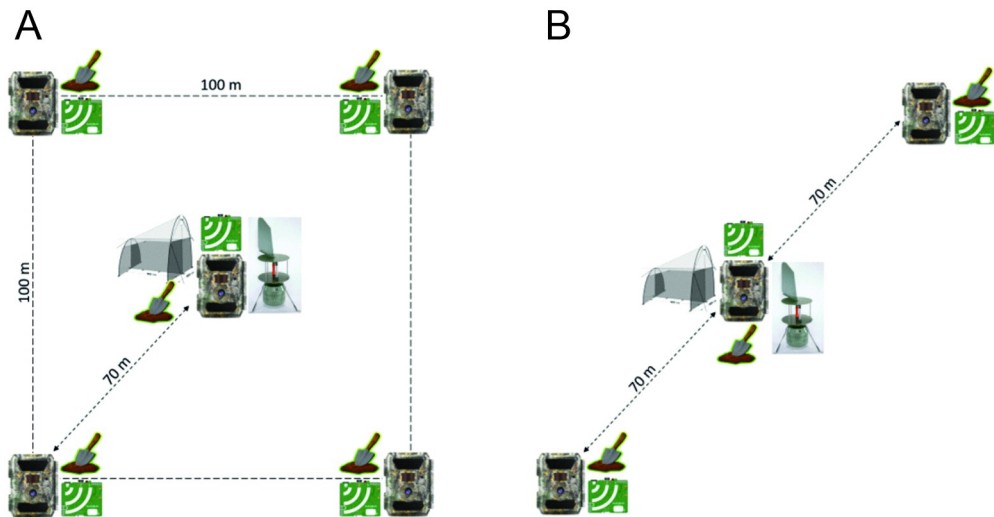

**Fig 6. Two alternate LIFEPLAN site designs.** (A) Design of LIFEPLAN one-hectare site with five sampling points. (B) Design of LIFEPLAN transect with three sampling points. Cameras and AudioMoths are placed within 10 m of each sampling point. Soil samples are collected in an area within approximately 5 m of the camera and audio recorder. The Malaise trap and cyclone sampler can be placed anywhere within the site, preferably close to the centre or middle but not in the camera's field of view.

## Sampling methods

Cameras and audio recorders collected data continuously year-round with weekly visits needed to change batteries and collect the data manually.

**Camera.** The protocol described in this peer-reviewed article is published on protocols.io, dx.doi.org/10.17504/protocols.io.q26g7pxp1gwz/v3 and is included for printing as S1 File with this article.

**Audio.** The protocol described in this peer-reviewed article is published on protocols.io, **dx.doi.org/10.17504/protocols.io.kqdg3xbp1g25/v2** and is included for printing as S2 File with this article.

Two 24-hour cyclone samples are collected while weather conditions allow the sampler to work, by visiting the site on three consecutive days each week or by using a timer for one of the samples and visiting the site two consecutive days per week. In a few cases, only one sample is collected using a timer due to the difficulty of reaching the site multiple days per week. The protocol described in this peer-reviewed article is published on protocols.io, dx.doi.org/10.17504/protocols.io.6qpvr3zn2vmk/v1 and is included for printing as S3 File with this article.

Flying arthropod samples are collected from Malaise traps once per week during the insect flying season producing a seven-day sample. The protocol described in this peer-reviewed article is published on protocols.io, dx.doi.org/10.17504/protocols.io.kqdg3xkdqg25/v2 and is included for printing as S4 File with this article.

One soil sample from each of three or five points defined by the camera and audio recorder locations is taken eight or four times per year when the temperatures are above freezing. The original frequency of eight times a year was reduced to four times a year in January 2023. The protocol described in this peer-reviewed article is published on protocols.io, dx.doi.org/10.17504/protocols.io.5jyl8pw9rg2w/v2 and is included for printing as S5 File with this article.

## Metadata

When collecting spatiotemporal data, it is essential to keep track of where and when each sample was collected. At a minimum, each sample must be associated with information on the date and time of sample start and end, and the sampling location.

We give each sample a unique identifier, which is a six-character code. A "sample" is either: a bottle containing a Malaise sample; a vial containing a cyclone sample; a bag containing a soil sample; or, for image and audio data, the contents of a memory card. For image and audio data, the uniquely identified memory cards are reused week after week and a sample is defined as one batch of data collected between memory card changes, so the unique identifier of an image or audio sample consisted of the six-character code of the memory card combined with the date on which the sample is collected.

We also give unique six-character codes to the sampling equipment. All equipment and samples are labelled with durable stickers that contain the unique identifier both as a human readable code and a scannable QR code.

When placing or collecting a sample, we use the LIFEPLAN mobile application [44] to scan the QR code of the equipment and the QR code of the sample. In the case of soil samples, we scanned only the sample code as there is no fixed equipment. This scanning creates an Activity, which is a row in a metadata table (Table 1).

**Table 1. Metadata fields recorded for each activity.**

| Field | Description | DarwinCore term |
| --- | --- | --- |
| Activity ID | Running number | N/A |
| Activity type | Selected by user: Placement (start of sample) / Collection (end of sample) | dwc:eventType |
| Activity UUID | Universally Unique IDentifier, e.g. 62185720-d7b4-4417-9385-d0bd072be2f8 | dwc:eventMeasurementID |
| Equipment type | Selected by user: Cyclone sampler / Malaise trap / camera trap / audio recorder / blank for soil samples | dwc:measurementMethod |
| Date and time | From mobile phone local date and time converted to UTC | dwc:eventDate |
| GPS latitude | From mobile phone location data, in decimal degrees | dwc:decimalLatitude |
| GPS longitude | From mobile phone location data, in decimal degrees | dwc:decimalLongitude |
| Equipment code | E.g. ET5J3G. Scanned from the QR code on the equipment. | N/A |
| Sample code | E.g. GDL2EM. Scanned from the QR code on the container or memory card. | dwc:materialSampleID |
| Site ID | Running number unique to a Site. Entered by the application from the database based on the equipment code, which is associated with a particular Site. | dwc:locationID |
| Team ID | Running number unique to a Team. A Team is in charge of both a natural and urban Site in the Global design. Entered by the application from the database based on the Equipment code, which is associated with a particular Site, which is associated with a Team. | N/A |
| User ID | Entered by the application from the database. All users are logged in with a personal username and password. | dwc:measurementDeterminedBy |
| Equipment condition | Selected by the user from a drop-down menu. | N/A |
| Sample condition | Selected by the user from a drop-down menu. | dwc:measurementRemarks |
| Notes | Open text field. | dwc:eventRemarks |
| Created at | Date and time that the Activity was created. | N/A |
| Updated at | Date and time that the Activity was last edited. | N/A |
| Deleted at | Date and time that the Activity was deleted. All deletions are soft deletions, where the Activity remains in the database but is marked deleted. | N/A |
| Link | Link to audio or image data: here users can paste a share link which gives access to the actual audio or image data stored in a cloud. After the data have been transferred to LIFEPLAN, the link becomes a link to the LIFEPLAN data repository. | N/A |
| Processing status | For image and audio data, this field indicates whether or not the data have been transferred to LIFEPLAN. | N/A |

For cases where the mobile application fails or metadata needs to be corrected, we also have a browser version: LIFEPLAN Web Admin [45]. Because metadata can be edited after collecting or created after the fact, we also record the fields described in Table 1.

For each sample, we thus hold spatiotemporal metadata of where and when it was placed in the field and collected. Our database has additional tables with information on the Sites, Teams, Equipment, Sample codes and Users, which we can associate with the Activity by the relevant IDs.

## Ethics declarations

The acquisition and transport of biological samples is strictly regulated by international agreements. All sampling and processing of samples within LIFEPLAN adheres to both national and international legislation and agreements.

To comply with the Convention on Biological Diversity, with each team contributing to LIFEPLAN we have a Material Transfer Agreement (MTA). With each nation, we have established via the ABS clearing house whether Prior Informed Consent (PIC) is required, and acquired it where necessary. The MTA and PIC define Access and Benefit Sharing clauses to provide non-monetary benefits to the data providers.

This study was subject to review by European Research Council Executive Agency (ERCEA) Unit B1 –Ethics Review and Expert Management.

## Expected results

### Samples for DNA based identification

Physical samples can be identified with metabarcoding, using different primers depending on the species group of interest. Malaise samples can also be individually sequenced or barcoded and photographed. In LIFEPLAN, we metabarcode all physical samples for the *COI* (cytochrome oxidase I) region for Malaise samples or *ITS2* (internal transcribed spacer 2) region for cyclone and soil samples, as well as individually *COI*-barcoding and bulk imaging a subset of Malaise samples.

Cyclone samples are collected in 1.5 ml sterile microtubes, then freeze dried and stored at -20˚C. They usually contain a small amount of fungal spores, pollen and other particles under 1 μm in diameter. Samples may contain small insects or water, which are removed and their presence noted before analysis (Fig 7A; Table 2).

Malaise samples are collected in Nalgene bottles filled with pure ethanol to euthanise and conserve the arthropods, and stored at -20 C˚ (Fig 7B).

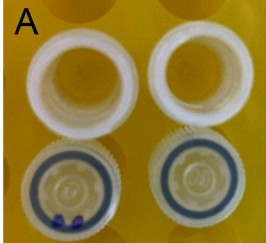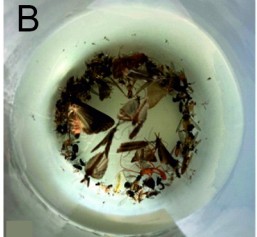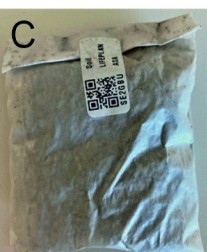

**Fig 7. Physical samples collected in project LIFEPLAN.** (A) Two cleaned and freeze-dried cyclone samples; an insect was removed from the tube on the left. (B) A Malaise sample collected after one week in the field. (C) A soil sample inside a glassine bag.

**Table 2. Example of a processing metadata table for cyclone samples.**

| Sample | Team | Site | Placement | Collection | Duration | Notes | Equipment condition | Sample field condition | Insect | Water | Freeze-dried |
|--------|------|------|-----------|------------|----------|-------|---------------------|------------------------|--------|-------|--------------|
| CWCBAW | SWE | SH11 | 7.10.2021 9:00 | 8.10.2021 8:58 | 0d 23h 58 min | | battery died | ok | | X | X |
| C4LY21 | SWE | UA21 | 11.10.2021 8:55 | 12.10.2021 08:56 | 1d 0h 1min | | ok | insect | X | | X |
| C6T1SX | AUS | AU4_Urban | 11.10.2021 3:10 | 12.10.2021 3:11 | 1d 0h 1min | | ok | ok | | | X |

Soil samples are collected in 75 x 102 mm glassine bags (Fig 7C). They contain a compound sample of about three tablespoons of soil. Samples are dried immediately after collection, then freeze dried and stored at -20˚ C until analysis.

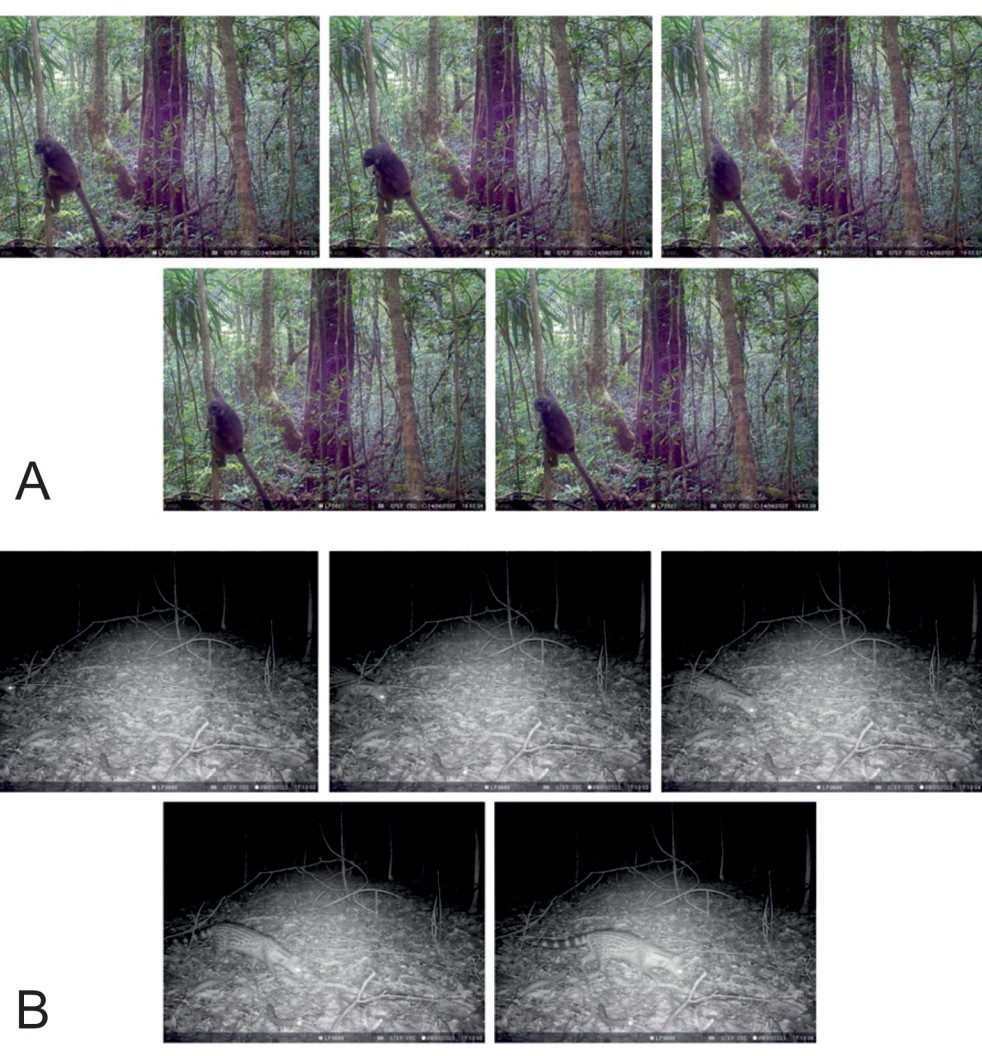

**Fig 8. Example of photo series from LIFEPLAN cameras.** The cameras record a five-image burst when triggered. (A) In daylight. (B) In darkness.

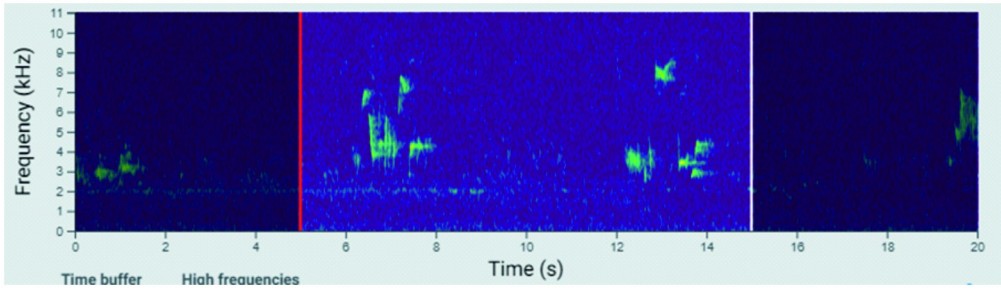

**Fig 9. Audio recording converted into a spectrogram for analysis.**

## Samples for automated species identification

Image data is in JPG format, in sequences of five images taken in a burst, approximately 1 second apart. The camera is triggered by movement, and images are taken both during the day and at night (Fig 8).

Audio recording data consist of regular one-minute segments, 48-hour continuous recordings, and opportunistic recordings in between regular segments that are triggered by bat calls. Audio recordings are converted to WAV format for analysis. Animal species can be recognised from spectrogram images of sounds by machine learning algorithms (Fig 9; [9]), as can camera trap images The LIFEPLAN automatic species identification software will be published elsewhere and made openly accessible.

## Supporting information

**S1 File. Camera trapping protocol.**
(PDF)

**S2 File. Audio recording protocol.**
(PDF)

**S3 File. Cyclone sampling protocol.**
(PDF)

**S4 File. Malaise trapping protocol.**
(PDF)

**S5 File. Soil sampling protocol.**
(PDF)

## Acknowledgments

Open Acoustic Devices provided the published version of the Lifeplan AudioMoth firmware. Pekka Niittynen calculated HFI and elevations for sites.

## Author Contributions

**Conceptualization:** Tomas Roslin, Otso Ovaskainen.

**Data curation:** Bess Hardwick, Deirdre Kerdraon.

**Funding acquisition:** Tomas Roslin, Otso Ovaskainen.

**Methodology:** Bess Hardwick, Deirdre Kerdraon, Hanna M. K. Rogers, Dimby Raharinjanahary, Eric Tsiriniaina Rajoelison, Tommi Mononen, Petteri Lehikoinen, Gaia Banelyte, Arielle Farrell.

**Project administration:** Bess Hardwick, Deirdre Kerdraon, Brian L. Fisher.

**Supervision:** Brian L. Fisher, Tomas Roslin, Otso Ovaskainen.

**Writing – original draft:** Bess Hardwick, Deirdre Kerdraon.

**Writing – review & editing:** Hanna M. K. Rogers, Dimby Raharinjanahary, Eric Tsiriniaina Rajoelison, Tommi Mononen, Petteri Lehikoinen, Gaia Banelyte, Arielle Farrell, Brian L. Fisher, Tomas Roslin, Otso Ovaskainen.

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
