## [Decision Letter · Decision Letter 0]

10 Sep 2024

PONE-D-24-09815LIFEPLAN: A worldwide biodiversity sampling designPLOS ONE

Dear Dr. Kerdraon,

Thank you for submitting your manuscript to PLOS ONE. After careful consideration, we feel that it has merit but does not fully meet PLOS ONE’s publication criteria as it currently stands. Therefore, we invite you to submit a revised version of the manuscript that addresses the points raised during the review process.

We look forward to receiving your revised manuscript.

Kind regards,

Bernd Schierwater, Ph.D

Academic Editor

PLOS ONE

 Please ensure that your manuscript meets PLOS ONE's style requirements, including those for file naming. The PLOS ONE style templates can be found at  https://journals.plos.org/plosone/s/file?id=wjVg/PLOSOne_formatting_sample_main_body.pdf and  https://journals.plos.org/plosone/s/file?id=ba62/PLOSOne_formatting_sample_title_authors_affiliations.pdf.   2. Please include a complete copy of PLOS’ questionnaire on inclusivity in global research in your revised manuscript. Our policy for research in this area aims to improve transparency in the reporting of research performed outside of researchers’ own country or community. The policy applies to researchers who have travelled to a different country to conduct research, research with Indigenous populations or their lands, and research on cultural artefacts. The questionnaire can also be requested at the journal’s discretion for any other submissions, even if these conditions are not met.  Please find more information on the policy and a link to download a blank copy of the questionnaire here: https://journals.plos.org/plosone/s/best-practices-in-research-reporting. Please upload a completed version of your questionnaire as Supporting Information when you resubmit your manuscript.   3. We note you have not yet provided a protocols.io PDF version of your protocol and/or a protocols.io DOI. When you submit your revision, please provide a PDF version of your protocol as generated by protocols.io (the file will have the protocols.io logo in the upper right corner of the first page) as a Supporting Information file. The filename should be S1_file.pdf, and you should enter “S1 File” into the Description field. Any additional protocols should be numbered S2, S3, and so on. Please also follow the instructions for Supporting Information captions [https://journals.plos.org/plosone/s/supporting-information#loc-captions]. The title in the caption should read: “Step-by-step protocol, also available on protocols.io.”   Please assign your protocol a protocols.io DOI, if you have not already done so, and include the following line in the Materials and Methods section of your manuscript: “The protocol described in this peer-reviewed article is published on protocols.io (https://dx.doi.org/10.17504/protocols.io.[...]) and is included for printing purposes as S1 File.” You should also supply the DOI in the Protocols.io DOI field of the submission form when you submit your revision.   If you have not yet uploaded your protocol to protocols.io, you are invited to use the platform’s protocol entry service [https://www.protocols.io/we-enter-protocols] for doing so, at no charge. Through this service, the team at protocols.io will enter your protocol for you and format it in a way that takes advantage of the platform’s features. When submitting your protocol to the protocol entry service please include the customer code PLOS2022 in the Note field and indicate that your protocol is associated with a PLOS ONE Lab Protocol Submission. You should also include the title and manuscript number of your PLOS ONE submission.   4. Your ethics statement should only appear in the Methods section of your manuscript. If your ethics statement is written in any section besides the Methods, please move it to the Methods section and delete it from any other section. Please ensure that your ethics statement is included in your manuscript, as the ethics statement entered into the online submission form will not be published alongside your manuscript. 

Reviewers' comments:

Reviewer's Responses to Questions

**Comments to the Author**

1. Does the manuscript report a protocol which is of utility to the research community and adds value to the published literature?

Reviewer #1: Yes

Reviewer #2: Yes

2. Has the protocol been described in sufficient detail?

To answer this question, please click the link to protocols.io in the Materials and Methods section of the manuscript (if a link has been provided) or consult the step-by-step protocol in the Supporting Information files.

The step-by-step protocol should contain sufficient detail for another researcher to be able to reproduce all experiments and analyses.

Reviewer #1: Yes

Reviewer #2: Partly

3. Does the protocol describe a validated method?

Reviewer #1: Yes

Reviewer #2: Yes

4. If the manuscript contains new data, have the authors made this data fully available?

Reviewer #1: N/A

Reviewer #2: Yes

**5. Is the article presented in an intelligible fashion and written in standard English?**

Reviewer #1: Yes

Reviewer #2: Yes

6. Review Comments to the Author

Reviewer #1: The manuscript by Hardwick et al. describes the sampling design of the LIFEPLAN project. I think it is good that the authors share project protocols as this will help other projects to generate data that can be compared with the data that was collected in LIFEPLAN project. This is of course going to be essential as continued and extensive efforts are needed in order to document and monitor the biodiversity in the planet.

On the contrary, in reviewing this manuscript, my main struggle has been that the sampling has been already carried out so I am not sure what a reviewer can feedback other than language, structure and minor details. Future publications of LIFESPAN could cite this study for easy description of the methods but this could have been a preprint as the reviewer serves little purpose but to ask questions about the past. I personally feel that an effective review of this sampling strategy can be done only in context of ecological conclusions and requires the reviewer to assess if the ecological conclusions of the study can be made/supported by this design. This aside, I will use this chance to help ask questions that other studies may ask so that this can be documented by the authors, along with a few minor comments.

1. It would be good to give readers an overview of LIFEPLAN in terms of when the project started, with its aims (“a central aim” is given in L83 but not others) so as to draw the interest of a general reader who is not aware of the project. It would also be good to share when the sampling was carried out, so it is easier for readers to make sense of the tense used in the introduction. There is mention of 2020 in methods (L244) but it comes too late.

2. L158-159: This statement raises many questions: similar geographic area with distinct evolutionary history can be said of many locations. it would be good if authors could suggest what kind of questions can be addressed by comparisons of these two sites, even if broadly, and how can these be robustly quantified when there are only 2 regions studied? The latter is of course understandable given this itself requires a huge scale of sampling.

3. L219: Why is there this difference of triangular and square design in Sweden and Madagascar? What would be recommended for future studies?

4. L284: is LIFEPLAN mobile app/LIFEPLAN Web Admin available for other researchers who would be interested in adopting the described protocols? I tried the app and the website but there is no way to “register”. Alternatively, is there some time in future this will be made available?

5. Table 1/metadata: Is there any plan to convert this to Darwin Core terms and keep it standardized such that these data can be also shared, for example to GBIF and in future be machine readable? The authors have motivated the manuscript by saying that we need consistent data (L83), and one key aspect of this is metadata consistency across studies. Ideally this should be done by following existing standards, unless these do not work for the research.

6. Figure 3B, There is a lone point near bottom left, it is not clear why that is present?

7. Figure 5, (A) square plot shows a distance of 50km, but the (B) distances in some cases look to me >100km. This could be due to difficulty in finding such sites but I think it would be good to mention ranges involved or state the issue.

8. L335: Figure 9 and citation (10) are both about audio data, not images.

Reviewer #2: I have reviewed the manuscript titled “LIFEPLAN: A worldwide biodiversity sampling design” and I found it useful as well as interesting. As a biodiversity sampling design, I think this is timely and can be used broadly worldwide.

The only major comment I have is that the introduction doesn’t seem to address the main topic and the reader is not really “introduced” to the context. For example, the authors highlighted current issues with materials from long-standing process not being identified on time, and yet they are presenting another sampling design and not an identification method. By reading the current introduction, one could argue we don’t need additional sampling methods if we still have decades of samples to identify.

Obviously I agree this work is useful, and the authors should make it more clear in the introduction.

Another example, at lines 73-75, this sentence may be a bit underappreciative of taxonomical efforts,

First of all, if we are talking about species identification only, this is an activity mostly performed by diagnosticians, who identify species (that have been described and discovered by others). The reason taxonomists are focused on smaller groups is due to the fact they also have to perform the species description, which is extremely time-consuming. This should be made abundantly clear in the introduction since your work focuses on collecting more specimens that will require taxonomists to study them if we want species to be identifiable.

I would suggest to replace the term “taxa” with “groups”, to make it clear you are not referring to species but larger taxonomical groups (e.g., families, superfamilies). Additionally, I would replace this sentence “and one human has a finite capacity for classification work” with something more on the line that we don’t have enough taxonomists available to cover all the taxonomical work required (this is a useful reference discussing this issue: https://doi.org/10.1093/zoolinnean/zlab072).

Overall, I would strongly suggest to redraft the introduction focusing on:

- Why do we need more standardised collection methods.

- What this method offers that other methods don’t.

- A bit more discussion on whether this new method could use data from other methods. As it is more and more important to have long-standing studies to record biodiversity, one could argue that yet another method (also quite expensive) may not be the best choice if instead we could adapt other methods/protocols that have already been implemented in the past few decades.

Minor suggestions/corrections:

Line 50: You should be more consistent throughout the MS and decide if you want to use the capital letter (line 178) or not (line 50) for “national” and “nested”. Since you are using these terms as “objects” of your analysis, I would suggest to use a capital letter but to define the concept here.

Lines 75-78: Following to my comment above, please, ensure the references mentioned here [7-8] did not also described new species, as there is a clear difference between species identification and species delimitation.

Line 123: Perhaps replace “foci” with “targets”, as the former is not really commonly used.

Line 154: Could you please list the Nordic countries? Even in brackets after the first mention.

Line 158: I think this is the best place to explain a bit better what you mean with “national”, “nested”.

Line 171: either remove capital letter from “Natural” or make “Urban” the same.

Line 192: not exactly sure what you mean with “realised” here. Could you please clarify or use a synonym?

Line 223: Replace “have” with “had” for consistency.

Line 226: Apologies, but I still don’t understand the meaning of “realised” in this context.. See comment above.

Line 266: “growing season” of what?

Line 305: Please, italicise all genes’ names.

Line 309: Add degree symbol.

7. PLOS authors have the option to publish the peer review history of their article (what does this mean?). If published, this will include your full peer review and any attached files.

Reviewer #1: No

Reviewer #2: No

---

## [Author Response · Author response to Decision Letter 0]

11 Oct 2024

Dear Dr Bernd Schierwater,

Thank you very much for your kind and careful evaluation of our manuscript (nr [PONE-D-24-09815] - [EMID:5279dfe19d786087], title LIFEPLAN: A worldwide biodiversity sampling design). We have done our very best to revise the manuscript so that it fully fits the journal requirements. In response to the valuable comments proffered on our revision, we have now made a series of changes. In particular, we have clarified the focus of the introduction and given the reasoning behind certain elements of the protocol. Below, we will detail the changes made and their justifications in a series of point-by-point responses. For clarity, we have included the original comments in italics and our responses in plain font.

 Given this extensive revision, we hope and trust that the manuscript is now ready for publication. Nonetheless, we remain committed to implement any further changes that you may feel needed.

On behalf of all authors,

NN

Response to Reviewers

LIFEPLAN: A worldwide biodiversity sampling design

Response: We have now carefully followed all style requirements, including those for file naming.

2. Please include a complete copy of PLOS’ questionnaire on inclusivity in global research in your revised manuscript.

Response: We have now included a filled-in copy of the questionnaire.

3. We note you have not yet provided a protocols.io PDF version of your protocol and/or a protocols.io DOI. When you submit your revision, please provide a PDF version of your protocol as generated by protocols.io (the file will have the protocols.io logo in the upper right corner of the first page) as a Supporting Information file. The filename should be S1_file.pdf, and you should enter “S1 File” into the Description field. Any additional protocols should be numbered S2, S3, and so on. Please also follow the instructions for Supporting Information captions [https://journals.plos.org/plosone/s/supporting-information#loc-captions]. The title in the caption should read: “Step-by-step protocol, also available on protocols.io.”

Please assign your protocol a protocols.io DOI, if you have not already done so, and include the following line in the Materials and Methods section of your manuscript: “The protocol described in this peer-reviewed article is published on protocols.io (https://dx.doi.org/10.17504/protocols.io.[...]) and is included for printing purposes as S1 File.” You should also supply the DOI in the Protocols.io DOI field of the submission form when you submit your revision.

If you have not yet uploaded your protocol to protocols.io, you are invited to use the platform’s protocol entry service [https://www.protocols.io/we-enter-protocols] for doing so, at no charge. Through this service, the team at protocols.io will enter your protocol for you and format it in a way that takes advantage of the platform’s features. When submitting your protocol to the protocol entry service please include the customer code PLOS2022 in the Note field and indicate that your protocol is associated with a PLOS ONE Lab Protocol Submission. You should also include the title and manuscript number of your PLOS ONE submission.

Response: All protocols have now been updated with their DOIs and attached as S1, S2, S3, S4, S5

Response: We have now moved our ethics statement to the Methods section.

Response: We have now carefully reviewed our list of references, making sure that it is complete and correct.

Reviewers' comments:

Reviewer #1: The manuscript by Hardwick et al. describes the sampling design of the LIFEPLAN project. I think it is good that the authors share project protocols as this will help other projects to generate data that can be compared with the data that was collected in LIFEPLAN project. This is of course going to be essential as continued and extensive efforts are needed in order to document and monitor the biodiversity in the planet.

On the contrary, in reviewing this manuscript, my main struggle has been that the sampling has been already carried out so I am not sure what a reviewer can feedback other than language, structure and minor details. Future publications of LIFESPAN could cite this study for easy description of the methods but this could have been a preprint as the reviewer serves little purpose but to ask questions about the past. I personally feel that an effective review of this sampling strategy can be done only in context of ecological conclusions and requires the reviewer to assess if the ecological conclusions of the study can be made/supported by this design. This aside, I will use this chance to help ask questions that other studies may ask so that this can be documented by the authors, along with a few minor comments.

Response: We appreciate the difficulty involved in reviewing a protocol which is de facto being implemented with specific questions in mind. In doing so, we note that the ecological questions asked and the conclusions reached will be the topic of separate papers to come (but see Abrego et al. 2024: Airborne DNA reveals predictable spatial and seasonal dynamics of fungi. Nature 631, 835-842. https://doi.org/10.1038/s41586-024-07658-9). Nonetheless, we find that the reviewer has invested great effort in clarifying the actual procedures, to the benefit of other researchers. This has helped us greatly improve the manuscript. We fully agree with the reviewer on the basic purpose of a protocol paper.

1. It would be good to give readers an overview of LIFEPLAN in terms of when the project started, with its aims (“a central aim” is given in L83 but not others) so as to draw the interest of a general reader who is not aware of the project. It would also be good to share when the sampling was carried out, so it is easier for readers to make sense of the tense used in the introduction. There is mention of 2020 in methods (L244) but it comes too late.

Response: We agree that this should be clarified and have added descriptions of the aims and timespan of LIFEPLAN sampling. 

2. L158-159: This statement raises many questions: similar geographic area with distinct evolutionary history can be said of many locations. it would be good if authors could suggest what kind of questions can be addressed by comparisons of these two sites, even if broadly, and how can these be robustly quantified when there are only 2 regions studied? The latter is of course understandable given this itself requires a huge scale of sampling.

Response: We note that the comparison between Madagascar and the Nordic countries will allow for a seemingly endless series of questions, of which we have specified some in the revised manuscript. Just how fruitful a comparison between two specific regions can be is illustrated by the findings from a parallel study focused on these two regions alone (van Dijk et al. 2024. Temperature and water availability drive insect seasonality across a temperate and a tropical region. Proc. B, 291, 20240090. https://doi.org/10.1098/rspb.2024.0090). Nonetheless, in the Lifeplan setting, we have made sure to avoid the unintended interpretation offered by the Reviewer. Lifeplan comparisons between geographic areas with distinct evolutionary history will naturally extend to all Lifeplan sites distributed globally, with the Nordic countries vs Madagascar as a case in point (given the hierarchical designs implemented there).

3. L219: Why is there this difference of triangular and square design in Sweden and Madagascar? What would be recommended for future studies?

Response: We have now added the reasoning behind the two different designs, as well as giving a recommendation for choosing between the two.

4. L284: is LIFEPLAN mobile app/LIFEPLAN Web Admin available for other researchers who would be interested in adopting the described protocols? I tried the app and the website but there is no way to “register”. Alternatively, is there some time in future this will be made available?

Response: Unfortunately the app and website will stop working when our funding ends, as keeping them online requires ongoing funding. We will make the code for the app and website publicly available so that other projects can make use of them. For this protocol paper, we aimed to describe what a metadata management system should record at a minimum, as well as describing how we achieved that with the very large volume of sampling within project LIFEPLAN.

5. Table 1/metadata: Is there any plan to convert this to Darwin Core terms and keep it standardized such that these data can be also shared, for example to GBIF and in future be machine readable? The authors have motivated the manuscript by saying that we need consistent data (L83), and one key aspect of this is metadata consistency across studies. Ideally this should be done by following existing standards, unless these do not work for the research.

Response: We have added a column to Table 1 giving Darwin Core terms where applicable. However strictly speaking, the metadata in Table 1 refer to Collection and Placement activities, not to actual samples or data. For instance, Table 1 has fields like the internal IDs of sampling equipment and datetimes when records were updated or soft-deleted. In these cases, Darwin Core terms are not applicable or the same term would have to be used multiple times, which is not allowed in Darwin Core. We agree that standardization via Darwin Core is essential for data sharing, and when we publish the actual data collected in LIFEPLAN we will be including sample-level metadata that complies with Darwin Core standards. 

6. Figure 3B, There is a lone point near bottom left, it is not clear why that is present?

Response: We thank the reviewer very much for pointing out this error. There were four sites that ended up only having a Natural location, as there were no populated areas close enough for an Urban type sampling point. Hence, these sites were also clustered at the very bottom of the figure with human footprint index values close to 0. These four sites should not have been included in the figure as they were not in fact paired sites, and have now been removed.

7. Figure 5, (A) square plot shows a distance of 50km, but the (B) distances in some cases look to me >100km. This could be due to difficulty in finding such sites but I think it would be good to mention ranges involved or state the issue.

Response: We have added explanations for the discrepancy between planned and realized distances, as well as the realized range of distances.

8. L335: Figure 9 and citation (10) are both about audio data, not images.

Response: “Images and sounds” should have read “Images of sounds”: this was an error. We have now further clarified that we mean spectrograms, which are the images that sounds are converted into for analysis.

Reviewer #2: I have reviewed the manuscript titled “LIFEPLAN: A worldwide biodiversity sampling design” and I found it useful as well as interesting. As a biodiversity sampling design, I think this is timely and can be used broadly worldwide.

The only major comment I have is that the introduction doesn’t seem to address the main topic and the reader is not really “introduced” to the context. For example, the authors highlighted current issues with materials from long-standing process not being identified on time, and yet they are presenting another sampling design and not an identification method. By reading the current introduction, one could argue we don’t need additional sampling methods if we still have decades of samples to identify. Obviously I agree this work is useful, and the authors should make it more clear in the introduction.

Response: We agree with Reviewer 2 that the introduction focused too much on the throughput challenges of morphological methods, when the method we are presenting here is really about generating big data. Processing these data will then take advantage of solutions developed and presented elsewhere – as being some of the main deliverables of methods development in Lifeplan. We have now shortened the discussion of throughput challenges to clarify the focus of the introduction.

Another example, at lines 73-75, this sentence may be a bit underappreciative of taxonomical efforts,

First of all, if we are talking about species identification only, this is an activity mostly performed by diagnosticians, who identify species (that have been described and discovered by others). The reason taxonomists are focused on smaller groups is due to the fact they also have to perform the species description, which is extremely time-consuming. This should be made abundantly clear in the introduction since your work focuses on collecting more specimens that will require taxonomists to study them if we want species to be identifiable.

Response: Any underappreciation of taxonomical efforts was most certainly unintended, and we have now edited the lines in question. We agree with the reviewer that framing the problem in terms of a lack of taxonomists is much more productive than pointing to an individual taxonomist’s capacity. The clarification of the terms “taxonomist” and “diagnostician” was a valuable addition, but as both are needed to produce biodiversity data, we have now opted for the more general term “expert”. We discuss the ramifications of the species identification problem in more detail in the penultimate paragraph of the introduction, and have now added a few words there to clarify that species identification and species description are indeed partly separate problems. Having said that, we note that accelerated procedures for species description are part of Lifeplan objectives, with separate papers to follow.

I would suggest to replace the term “taxa” with “groups”, to make it clear you are not referring to species but larger taxonomical groups (e.g., families, superfamilies). Additionally, I would replace this sentence “and one human has a finite capacity for classification work” with something more on the line that we don’t have enough taxonomists available to cover all the taxonomical work required (this is a useful reference discussing this issue: https://doi.org/10.1093/zoolinnean/zlab072

Response: We thank the reviewer for these suggestions and have made changes accordingly.

Overall, I would strongly suggest to redraft the introduction focusing on:

- Why do we need more standardised collection methods.

- What this method offers that other methods don’t.

- A bit more discussion on whether this new method could use data from other methods. As it is more and more important to have long-standing studies to record biodiversity, one could argue that yet another method (also quite expensive) may not be the best choice if instead we could adapt other methods/protocols that have already been implemented in the past few decades.

Response: We have now reframed the introduction. In doing so, we have emphasized that our methods do build directly on methods and protocols developed by others. Using a Malaise trap is no new innovation, but operating it a

---

## [Editor Report · Decision Letter 1]

23 Oct 2024

LIFEPLAN: A worldwide biodiversity sampling design

PONE-D-24-09815R1

Dear Dr. Kerdraon,

We’re pleased to inform you that your manuscript has been judged scientifically suitable for publication and will be formally accepted for publication once it meets all outstanding technical requirements.

Kind regards,

Bernd Schierwater, Ph.D

Academic Editor

PLOS ONE
---

## [Editor Report · Acceptance letter]

10 Dec 2024

PONE-D-24-09815R1 

PLOS ONE

Dear Dr. Kerdraon, 

I'm pleased to inform you that your manuscript has been deemed suitable for publication in PLOS ONE. Congratulations! Your manuscript is now being handed over to our production team.

Kind regards, 

on behalf of

Prof. Bernd Schierwater 

Academic Editor

PLOS ONE